# DFT Investigations of Au*_n_* Nano-Clusters Supported on TiO_2_ Nanotubes: Structures and Electronic Properties

**DOI:** 10.3390/molecules27092756

**Published:** 2022-04-25

**Authors:** Ying Wang, Ge Zhou

**Affiliations:** College of Chemistry, Sichuan University, Chengdu 610064, China; wy091213zyx@163.com

**Keywords:** DMOL3, TiO_2_-nanotube, Au*_n_*, geometries, electronic properties

## Abstract

TiO_2_ nanotubes (TiO_2_NTs) are beneficial for photogenerated electron separation in photocatalysis. In order to improve the utilization rate of TiO_2_NTs in the visible light region, an effective method is to use Au*_n_* cluster deposition-modified TiO_2_NTs. It is of great significance to investigate the mechanism of Au*_n_* clusters supported on TiO_2_NTs to strengthen its visible-light response. In this work, the structures, electronic properties, Mulliken atomic charge, density of states, band structure, and deformation density of Au*_n_* (*n* = 1, 8, 13) clusters supported on TiO_2_NTs were investigated by DMOL3. Based on published research results, the most stable adsorption configurations of Au*_n_* (*n* = 1, 8, 13) clusters supported with TiO_2_NTs were obtained. The adsorption energy increased as the number of Au adatoms increased linearly. The Au*_n_* clusters supported on TiO_2_NTs carry a negative charge. The band gaps of the three most stable structures of each adsorption system decreased compared to TiO_2_NTs; the valence top and the conduction bottom of the Fermi level come mainly from the contribution of 5*d* and 6*s*-Au. The electronic properties of the 5*d* and 6*s* impurity orbitals cause valence widening and band gap narrowing.

## 1. Introduction

The research on nanostructure models of metal clusters supported on well-ordered metal oxide surfaces is significant, providing important insights into the properties and mechanisms of real catalyst systems, and thus has been conducted extensively over the past few years [1,2,3,4,5,6,7,8,9,10,11,12,13]. In these model systems, the nature and strength of the interaction between the metal clusters and the support materials not only govern the nucleation and stability of the metal clusters, but also control the geometric and electronic structure of the resulting cluster/oxide interface, which are in turn critical to the catalytic activity of oxide-supported metal clusters [1,2]. Titanium dioxide has attracted worldwide attention due to its potential applications in a wide variety of products, such as photocatalysis [14,15], solar cells [16,17,18], and designing nanostructure architectures [19,20,21,22,23], due to its excellent gas sensitivity, moisture sensitivity, dielectric effect, photoelectric conversion and photocatalytic properties, chemical stability, nontoxicity, and relatively low cost. Among the three different polymorphs of TiO_2_, rutile, anatase, and brookite, the anatase phase has been extensively studied over the last few decades due to its technological applications and photocatalysis [24]. However, the low surface area (ca. 50 m^2^/g) and the large band gap of TiO_2_ (about 3.2 eV in the anatase phase [25]) limit its light absorption to only 5% of the solar spectrum [26,27,28], restricting its applications. In order to make more use of solar energy to increase the photocatalytic efficiency, it is more beneficial for absorbing visible light to reduce the band gap of TiO_2_ and increase the surface area of TiO_2_ materials. As is well known to us, among the nanostructures, namely simple assemblies of nanoparticles, one-dimensional (1D) nanostructures (nanorods, nanowires, and nanotubes) have a relatively large surface area. Accordingly, the TiO_2_ nanotubes (TiO_2_NTs) constructed by anatase are a promising structure with a large surface area of 328 m^2^/g [29]. Furthermore, TiO_2_NTs have a smaller band gap than the bulk powders and strong adsorption capacity, which will be desired to improve the photocatalytic properties and photoelectricity conversion effects, and have attracted considerable attention. A 3D random network of nanoparticles with more particle boundaries is more beneficial for transporting charge carriers than 1D nanostructures. Therefore, the collection of photogenerated charge carriers should be enhanced for adopted 1D nanostructure materials [30,31,32,33].

Although the smaller band gap of TiO_2_NTs was beneficial in the absorbing of visible lights [34], they clearly required more solar energy than modified nanotubes. In order to enhance the visible light sensitivity of TiO_2_ photocatalyst materials, many efforts have been made. The main modification methods are the organic dye photosensitive process, noble metal deposition, metal ion doping, the semiconductor coupling method, and so on. Metal nanoparticles (NPs) have considerably better photostability than semiconductor quantum dots, which usually suffer from anodic corrosion [26,35,36]. Some metal elements such as Au, Pt, Ag, Cu, and Fe have been employed to tune the electronic structure and enhance the catalytic activity of TiO_2_NTs [4,37,38,39,40,41,42,43,44]. Among various metal clusters, the Au cluster is considered as one of the most promising candidates owing to its high conductivity, good stability in the air, and controllability of the electrical properties. In contrast to the noble character of bulk Au, the Au cluster/metal oxide system (such as Au/TiO_2_) has been extensively studied due to the unique catalytic activity of this system. Recently, Zhi and co-workers [37] demonstrated that an Au nanoparticle–TiO_2_ nanotube junction was formed by using bovine serum albumin as a biotemplate, with the Au nanoparticles formed in the tube channels and even in the space between the tube bottom and the Ti substrate, which show highly improved electrochemical conductivity and act as electrode materials to achieve the enhanced direct electrochemistry of heme proteins. Shao et al. [38] reported plasmonic Au particles loaded on anodic TiO_2_ nanotube films exhibiting about 145% enhancement of the photocurrent and 37% reduction in response time. Chen et al. [39] synthesized the TiO_2_ nanotube-supported Pt/Au nanoparticles by means of the photo-assisted deposition approach. The studies have shown that TiO_2_NT-supported Pt/Au nanoparticles exhibit very high electrocatalytic activity toward formic acid oxidation when the Au composition is between 30% and 50%. Zhao et al. [4] prepared Au (or Pt) loaded on TiO_2_ nanotubes by means of the photo-deposition method for the degradation of methyl orange. The Au-loaded sample with an adsorption peak in the visible range becomes a visible light photocatalyst. In addition, other researchers [40,41,42,43,44] combined Au nanoparticles with other metal-loaded TiO_2_NTs to achieve high catalytic performance.

For Au nanoparticles supported on TiO_2_ nanotube systems, numerous investigations have been performed through many experiments. By contrast, the theoretical research providing important insights into the properties and mechanisms of real catalyst systems is scarce and significant. The purpose of this work is to investigation the micromechanisms of energy and charge transport in the nano-junction consisting of TiO_2_NTs and Au*_n_* clusters by building a nanostructure catalyst model. In this work, we report our results of first principles DFT calculation of Au*_n_* clusters supported on the anatase TiO_2_NT surface. The TiO_2_NTs were obtained by rolling up a TiO_2_ (101) surface, which was prepared by cleaving anatase bulk TiO_2_ perpendicular to the [1 0 1] direction. The anatase (101) surface was selected due to its thermodynamic stability [45,46,47,48]. In adsorption experiments, we adopted a single Au atom to test the adsorption sites of the TiO_2_NT surface. For the catalytic performance of Au, nanoparticles were defined by three major factors: contact structure, support selection, and particle size [49]. The 3D conformations are better suited for adsorption on a surface and concomitant oxidation and reduction reactions compared with planar conformations [7,50]. The measurements of CO oxidation using the temperature-programmed reaction by A. Sanchez [51] showed that the smallest gold cluster that catalyzes the reaction is Au_8_. Similar results were obtained by Hannu Häkkinen [52] and Bokwon Yoon [53]. Based on previous research [49,50,51,52,53,54], the adsorptive conformation of Au_8_ biplanar was adopted in our study. Atomic clusters show both electronic and geometric magic numbers, and 13 is a common magic number for many transition-metal clusters, including Au [55,56]. For Au*_n_* nanoclusters with 11 to 14 atoms, there appears to be a transition from 2D to 3D structures [57]. The work of J. Oviedo suggested that the most stable structure of Au_13_ comprises face-centered cubes or icosahedrons [55]. Ghazal Shafai et al. [57] selected the lowest-energy isomers for four types of cluster: planar, flake, cuboctahedron, and icosahedrons. The results show that there is no energy barrier between the icosahedron and the cuboctahedron configurations. Under the principle of three-dimensional configuration, we adopted a face-centered cubic and icosahedron structure for the original structures of the Au_13_ cluster. After optimization, the icosahedrons deformed into a distorted face-centered cubic structure, which is consistent with results of [57]. Therefore, we used a cuboctahedron as the initial configuration of the Au_13_ cluster. After the geometric optimization without symmetry being restricted, an energy-stable Au_13_ cuboctahedral structure was obtained. In order to understand the driving mechanism that determines the morphology and charge transport of TiO_2_NT-supported Au*_n_* nanoparticles, the structural and electronic properties of the adsorption systems were studied.

## 2. Methodology

The geometric structures of the bare nanotube and bare Au*_n_* clusters are shown in Figure 1. All the calculations were performed using the semi-core pseudopotential method within the DFT framework. Exchange and correlation terms were considered within the generalized gradient approximation (GGA) with a Perdew–Burke–Ernzerof (PBE) functional [58], the all-electron double numerical basis set with a polarized function (DNP), as implemented in Dmol3 code [59,60]. A tetragonal supercell with the size of 40 Å × 40 Å × c Å was set, where the parameter c was 11 Å, equal to the minimum periodic unit length of the TiO_2_NT (6,0). The supercell included 32 titanium and 64 oxygen atoms with the crystal form of (TiO_2_)_32_. The Brillouin zone was sampled by 4*4*2 [61] special k-points using the Monkhorst Pack [62] scheme for geometrical optimizations and the electronic properties calculation of TiO_2_ anatase, TiO_2_NT, and adsorption systems, respectively. A spin-restricted formalism was employed even in the presence of unpaired electrons, as the geometrical optimization is extremely sensitive to the details of the computational approach. The calculated bulk anatase TiO_2_ lattice parameters (*a* = *b* = 3.8283 Å, *c* = 9.5734 Å, *u* = 0.2080 Å, where *u* = *d*_ap_/*c* is the internal coordinate and *d*_ap_ is the Ti-O top bond length) agree well with the experiments [25,63]. The calculated band gap of pure anatase TiO_2_ is 2.77 eV, which is smaller than the experimental value, 3.2 eV [25]. This is due to the fact that density functional theory does not consider the electronic exchange-correlation potential discontinuity, which results in the basic band gap width being smaller than the experimental value by about 30–50%, generally. This does not affect the analysis of the electronic structure. The initial single-walled anatase TiO_2_NT models were constructed by rolling up one (101) layer of the anatase structure in the $\left[\bar{1}01\right]$ direction [30,34,64,65]. The (101) layer has 12 atoms (four titanium and eight oxygen atoms) in the unit cell and with the basic vectors V and U in the [010] and $\left[\bar{1}01\right]$ directions, respectively. The nanotubes were obtained by rolling up the layer in ways in which the chiral vectors (6,0) = 6 V became the circumferences of the nanotube. The 1D line symmetry group of the nanotube TiO_2_NT (6,0) can be represented as P42/mmc (D4H-9).

According to the work of Vittadini and Selloni on TiO_2_ (101) surface adsorption of Au clusters [3], the adsorption energies for Au*_n_* clusters are as follows:(1)$$E_{{\mathrm{Au}}_{n}}^{\mathrm{ads}}=-\left(E_{{\mathrm{Au}}_{n}/{\mathrm{TiO}}_{2}\mathrm{NT}}-E_{{\mathrm{TiO}}_{2}\mathrm{NT}}-E_{{\mathrm{Au}}_{n}}\right)$$
where $E_{{\mathrm{Au}}_{n}/{\mathrm{TiO}}_{2}\mathrm{NT}}$ ($E_{{\mathrm{TiO}}_{2}\mathrm{NT}}$) represents the energy of the nanotube with (without) the adsorbate, and $E_{{\mathrm{Au}}_{n}}$ denotes the energy of the gas-phase cluster.

We also define a cohesive energy to obtain information about the clustering energetics:(2)$$E_{{\mathrm{Au}}_{n}}^{\mathrm{clu}}=-\left(E_{{\mathrm{Au}}_{n}/{\mathrm{TiO}}_{2}\mathrm{NT}}-E_{{\mathrm{TiO}}_{2}\mathrm{NT}}-nE_{\mathrm{Au}}\right)/n$$
where *E*_Au_ is the total energy of a free Au atom. $E_{{\mathrm{Au}}_{n}}^{\mathrm{ads}}=E_{{\mathrm{Au}}_{n}}^{\mathrm{clu}}$, when a single Au adatom is on the TiO_2_NT surface.

Electronic structure analyses, including Mulliken charge and density of states (DOS), the partial density of states (PDOS), as well as deformation density, energy gap, and molecular orbital, were performed with DMOL3 of Materials Studio package (MS, version 8.0 Accelrys Software Inc., San Diego, CA, USA). These analyses were used to help us understand the nature of bonding and the interaction between Au*_n_* clusters and anatase TiO_2_NTs.

## 3. Results and Discussion

### 3.1. Structures of Anatase TiO_2_ Nanotubes

The cross-sectional and side view of the optimized TiO_2_NT (6,0) are shown in Figure 1. In the TiO_2_NT (6,0), both the inner and outer walls were terminated with the two-fold-coordinated oxygen atoms (2cO). In addition to 2cO atoms, three-fold-coordinated oxygen atoms (3cO) as well as five-fold-coordinated (5cTi) atoms are also exposed on the surface of the TiO_2_NT (6,0).

### 3.2. Structures of Au_1_/TiO_2_NTs

Two different stable adsorption structures were found for a single Au adatom on the TiO_2_NT (6,0) surface, as shown in Figure 2: a symmetric bridging site between two edge 2cO atoms in the $\left[\bar{1}01\right]$ direction, Au_1(O,O)_, and the other right on top of a 3cO atom, as well as bonding to three 5cTi atoms, Au_1(O,Ti)_. The adsorption energies of Au in these two configurations are listed in Table 1. For the two stable configurations, the adsorption energy of Au_1(O,Ti)_ is 0.49 eV. Au_1(O,Ti)_ is significantly more stable than Au_1(O,O)_, which has an adsorption energy of about 0.20 eV. To illustrate the charge of adsorbed clusters and the charge distributions of clusters and related TiO_2_NT surface atoms, Mulliken charge analysis was employed. The Mulliken charges of Au and the nanotube surface atoms that directly associated with the Au adatom are shown in Table 1. Apparently, Au became negatively charged by receiving electrons in both configurations. Both types of O atoms binding to Au directly became less negative, and most 5cTi atoms became more positive, except for 5cTi^3^, which generally indicated the loss of electrons. This is consistent with Au adsorption on the anatase TiO_2_ (101) surface [3].

The DOS plots of bare TiO_2_NT and the adsorption system, and the PDOS plots of the Au adatom and the oxygen atoms and titanium atoms of TiO_2_ nanotube, are shown in Figure 3 and are used to further illustrate bonding characteristics. For the projection of 2*p* orbitals of oxygen, 3*d* orbitals of titanium and 5*d* orbitals of gold showed the major contribution to the PDOS in the energy range of interest. Therefore, projections of individual orbitals along with the DOS are shown. The zero on the energy axis of the plots corresponds to the Fermi level of the bare TiO_2_NT and is at the top of the valence band, marked with a red dashed line in the figures. As presented in Figure 3a, the bare TiO_2_NT is semiconducting, and the valence band mostly has a contribution by the O 2*p* orbital (red line in Figure 3a) with a small contribution from the Ti 3*d* orbital (blue line in Figure 3a). The conduction band is dominated by the Ti 3*d* orbital. The electron density contributed by the O 2*s* orbital in the low energy region from −0.8 Ha to −0.5 Ha has little effect on the electron structure regarding the Fermi level. Therefore, in the following parts, we do not discuss this any further. According to Figure 3b,c, after an Au atom was adsorbed on TiO_2_NTs, the TiO_2_NTs retained some semiconductor properties. The Fermi level moved close to the bottom of conduction band, and the valence band had mostly an O 2*p* character, but a new small peak of the Au 5*d* atom arose at the top of it. Correspondingly, a smaller peak of Au *6**s* appeared in the bottom of the conduction band at the zero of the energy axis. Compared with Au_1(O,O)_, the Au 5*d* peak arose in the valence band separately. The valence band of Au_1(O,Ti)_ contributed by the Au 5*d* peak, overlapped well with the O 2*p* and Ti 3*d*. It is thus more likely to cause electron transfer. The contribution of the Au 5*d* and 6*s* orbitals at the top valence band and bottom conduction band causes valence band broadening and band gap narrowing, which causes the adsorption band edge of TiO_2_NTs to red shift.

The highest occupied molecular orbital (HOMO) and the lowest unoccupied molecular orbital (LUMO) along with the energy gap of bare TiO_2_NTs and the adsorption system are shown in Figure 4. In the two adsorption configurations, the Au atoms are supported on the TiO_2_NT surface, reducing the energy gap from 2.704 eV to 1.979 eV and 2.592 eV, respectively. Further analysis shows that the *d* orbital of Au adatoms has an evident contribution to the HOMO orbital. The results are the same as the previous analysis of DOS. This leads to the energy of HOMO (Au_1(O,O)_) rising from −7.171 eV to −6.494 eV. Meanwhile, the major feature of HOMO is that the Au 5*d* orbital as well as the O 2*p* orbital replaced the main contribution of O 2*p*. The energy of LUMO (Au_1(O,O)_) decreases from −4.467 eV to −4.515 eV. The 3$d_{x^{2}-y^{2}}$ orbital of Ti gives the contribution to the bottom of the conduction band. The character of the 6*s* orbital of Au was not obviously observed in the LUMO orbital.

The bonding characteristics of the adsorption system are further demonstrated in the electron deformation density (EDD) contour maps in Figure 5. The EDD contour maps were defined as the total density by subtracting the isolated atoms’ electron density. Compared with the Au_1(O,O)_, which has no significant bonding between Au atoms, and 2cO atoms deposited on the TiO_2_NT surface, the Au_1(O,Ti)_ has an obvious depletion of electron density when the 3cO 2p orbital is aligned with the Au–3cO bond direction. Additionally, the Au atom was surrounded by a small number of electrons, which indicated the electron transfer from TiO_2_NTs to gold nano-clusters.

### 3.3. Structures of Au_8_/TiO_2_NT

Figure 6 shows four different configurations of Au_8_ clusters adsorbed on the TiO_2_NT surface. The structures of Au_8_ in the four adsorption systems still maintain the biplanar conformation. The adsorption energy and Mulliken charge analysis for all these structures are listed in Table 2. Au_8-A(2cO,3cO,2cO)_ is the most stable configuration for the Au_8_/TiO_2_NT systems, as shown in Figure 6(A-a,A-b). In this structure, three Au atoms as an adsorption layer bond to TiO_2_NT surface atoms, and the oxygen atoms of the TiO_2_NT surface which interacted with the Au_8-A_ were 2cO, 3cO, and 2cO, respectively. In the structure shown in Figure 6(B-a,B-b), Au_8-B(2cO,5cTi)_ has an adsorption energy of about 0.86 eV, slightly smaller than Au_8-A_, which has an adsorption energy of 1.11 eV. For the adsorption configuration of Au_8-B_, similar to Au_8-A_, there was an adsorption layer consisting of three Au atoms from the side direction of original Au_8_ biplanar binding to the TiO_2_NT directly. The third structure, Au_8-C(2cO,2cO,3cO)_, which is displayed in Figure 6(C-a,C-b), is almost as stable as Au_8-B_. The energy difference between the two structures is merely 0.05 eV. In Au_8-C_, the adsorption layer is made up by four Au atoms as the bottom layer connecting with the TiO_2_NT surface. Both the bottom and top layer are part of the rhombus. The fourth structure, Au_8-D(2cO,2cO,3cO)_, is a result of the relaxation by five Au atoms of Au_8_ biplanar parallel to the axial of TiO_2_NTs. Au_8-D_, as a local minimum, has similar binding sites to Au_8-C_, and is evidently less stable than the three previous structures. Noticeably, the adsorption of Au adatoms caused the TiO_2_NT surface’s deformation; the 3cO atoms which bonded with Au adatoms were pulled off of the surface of TiO_2_NT. Meanwhile, the other 3cO atoms which have no relation with the Au_8_ clusters were “pushed” into a slightly concave formation. This phenomenon can be found in Figure 6. According to Table 2, the Au_8_ clusters in four structures were negatively charged. Based on analysis of Au nanocluster size with Mulliken atomic charge and adsorption energy, for Au_8_ clusters, not only transfer charges but also adsorption energies were increased, with clusters enlarging. Compared to Au_1_ absorption systems with a charge of about −0.060 au and −0.075 au, respectively, there was a remarkable increase in the electron transfer to Au_8_ clusters. The maximum charge is −0.424 au for the Au_8-C_ structure. This phenomenon suggested that more charge transfer will be required to enlarge the size of Au*_n_* nanoclusters to a certain degree. The interatomic charge distributions of related TiO_2_NT surface atoms were analyzed in detail. Apart from a few TiO_2_NT surface atoms, there are two main trends in atomic charge redistribution after the adsorption of Au_8_ clusters: electron transfer to oxygen atoms, and titanium losing electrons and showing greater positive charge.

PDOSs of Au_8_ clusters as well as the associated TiO_2_NT surface atoms in all four adsorption structures are plotted in Figure 7. The mixing between the O 2*p* orbital and Au 5*d*6*s* states spans the whole energy range of the valence band. Compared with the PDOS of a single Au adatom, the intensity of Au_8_ clusters at the Fermi level was greatly increased. The Au clusters’ 6*s* states make a dominating contribution to the states for the gap and are closely related to the Fermi level. The fact is that the energy gap of bare TiO_2_NTs almost disappears in all four adsorption systems. This indicates that metallization of the nanojunction Au_8_-TiO_2_NT system occurs, and therefore further increases tunnelling currents [66].

For more detailed analysis, the molecular orbital diagrams of HOMO and LUMO along with band gaps of the adsorption systems are shown in Figure 8. As the Au_8_ clusters adsorb on the TiO_2_NT surface, the contribution of metal clusters to the HOMO and the LUMO of the Fermi level is increased. As in the analysis of DOS, the band gap of the Au_8_–TiO_2_NT nanojunction was below 1.10 eV; Au_8_ clusters narrowed the band gap of the system more efficaciously than a single Au adatom. From the EDD contour maps of the Au_8_ adsorption system in Figure 5e,f, it can be found that Au_8_ clusters obtained electrons from TiO_2_NTs. In addition to the electron cloud distribution in Au_8_ nanoclusters, the Au atoms which have direct bonding with TiO_2_NT surface atoms show obvious electron accumulation, such as a Au_8-C_ structure, as shown in Figure 5f.

### 3.4. Structures of Au_13_/TiO_2_NT

Au_13_ clusters in the gas phase have two stable 3D arrangements: icosahedrons and cuboctahedrons. There is no energy barrier between these two configurations. In our study, the icosahedrons deformed into a distorted face-centered cubic structure after geometry optimization. Au_13_ clusters possessing the more stable configuration of cuboctahedrons were selected to construct the Au_13_/TiO_2_NT system. Three stable adsorption configurations for Au_13_/TiO_2_NTs were obtained. The investigated configurations for the three adsorption systems are displayed in Figure 9. We can find that the condition of TiO_2_NT surface oxygen atoms which bind with the cluster is similar to the Au_8-A(2cO,3cO,2cO)_. For the Au_13-A(2cO,2cO)_ configuration, in which the cluster is parallel to the TiO_2_NT axial, Au_13_ bonded to two oxygen atoms on the nanotube surface and possessed the conformation with Au atoms spreading on the outside of nanotube surface, which is energetically preferred (see Table 3). In Au_13-B(2cO,3cO,2cO,3cO)_, the absorbed Au_13_ cluster is similar to a cage-like structure [67,68], with four Au atoms directly bonding to the TiO_2_NT surface. Since the cage-like structure covers less area of the TiO_2_ nanotube than the flat geometry does, the density of the interfacial sites of the former is less than that of the latter. Although the cage-like structure is slightly less energetically favored than one of the geometries considered here, when the interface is considered as the controlling parameter, the cage-like Au_13_ nanoclusters can be expected to be more active than the other one for catalytic reactions. For optimized Au_13-C(3cO,2cO)_ configuration, it also presents as a cage-like configuration. Au_13_ is adsorbed with a three-coordination oxygen atom and a two-coordination bridge oxygen atom, as shown in Figure 9C. The interaction of the Au_13_ with the nanotube surface was further analyzed with Mulliken charges. The adsorption energy for these structures and the Mulliken charges on the Au atoms and the binding atoms of TiO_2_NT are summarized in Table 3. Au_13_/TiO_2_NT systems have greater adsorption energy than Au_8_ and Au_1_ systems (see Table 3). This indicates that increasing the number of Au atoms can enhance the stabilization of the adsorption system and the size scope of Au clusters according to the results of this study for the Au nanoclusters. According to Table 3, with previous Mulliken charge analysis results of Au_1_ and Au_8_, the Au_13_ clusters became negative after being absorbed on TiO_2_NTs. However, the number of electrons transferred experiences no change. The interatomic charge distributions of related TiO_2_NT surface atoms are shown in Table 3. The two main trends of atomic charge redistribution after the adsorption of Au_13_ clusters are: electron transfer to oxygen atoms and titanium losing electrons and showing a more positive charge, except for a few TiO_2_NT surface atoms, which is similar to the Au_8_ system. For further details, we integrated Mulliken charge analysis with deformation density (see Figure 5g,h); noticeably, there are significant electrons missing on the three coordination oxygen atoms, while the connected Au atoms obtain electrons, which are highlighted using a red dashed line in the EDD contour maps in Figure 5h.

Figure 10 shows the PDOS of Au_13_ clusters and connecting surface atoms of TiO_2_NT along with the DOS of adsorption system. 5*d*-Au has a contribution to the top of the valence band and overlaps well with the O 2*p* orbital for all three adsorption structures. With the electron density of 5*d*6*s*-Au being raised, the contribution from Au atoms in the valence band and the conduction band was more and more obvious. The Au_13_ cluster adsorbed onto the TiO_2_NT surface will have an excellent performance in electron transport. The HOMO and LUMO orbital diagrams of adsorption systems, as shown in Figure 11, visually illustrate the orbital analysis results given the density of states. Au_13_ nanoclusters contribute to the valence band and the conduction band, along with the increase in the number of gold atoms. HOMO and LUMO orbitals are mainly provided by the Au nanoclusters. Compared with the Au_8_/TiO_2_NT system, the system’s energy is further decreased, particularly for the Au_13-A_ configuration, which downgraded the energy gap to 0.049 eV, far below the maximum energy value of the visible light absorption.

## 4. Conclusions

In summary, Au*_n_*/TiO_2_NTsystems are studied using density functional theory to characterize the effect of the adsorption of Au*_n_* (*n* = 1, 8, 13) clusters on the geometric and electronic structures of anatase TiO_2_NT. Our results show that a single Au adatom prefers the top position of 3cO as well as the bridging 2cO–2cO site of the TiO_2_NT surface. Adsorbed Au_8_ maintains a distorted biplanar configuration. The strong interaction between Au*_n_* and atoms of the nanotube surface causes deformation of TiO_2_NT. Au_13_ is adsorbed in a cage-like structure and has a tendency to spread out on the wetted nanotube surface.

The adsorption energy is increased as the number of Au atoms increases linearly, and increases in the size of Au*_n_* clusters are conducive to stabilizing the load systems. The peaks at the Fermi level of the valence top and the conduction bottom come mainly from the contribution of 5*d*6*s*-Au atoms. The 5*d*6*s* orbital of impurities’ electronic state density caused valence widening and band gap narrowing. The band gap of the three most stable structures of adsorption systems Au_1-A_ (2.592 eV), Au_8-A_ (1.100 eV), and Au_13-A_ (0.049 eV) decreased compared to bare TiO_2_NTs. This causes TiO_2_NTs to achieve a visible light response. Molecular orbital diagrams intuitively verify the obviously increasing contribution to HOMO and LUMO orbitals with the increase in gold atoms. Our present results serve as a possible indicator that the nanojunction TiO_2_NT/Au*_n_* cluster, as a potential photoelectric device, possesses better energy and charge transmission performance.

## Figures and Tables

**Figure 1 molecules-27-02756-f001:**
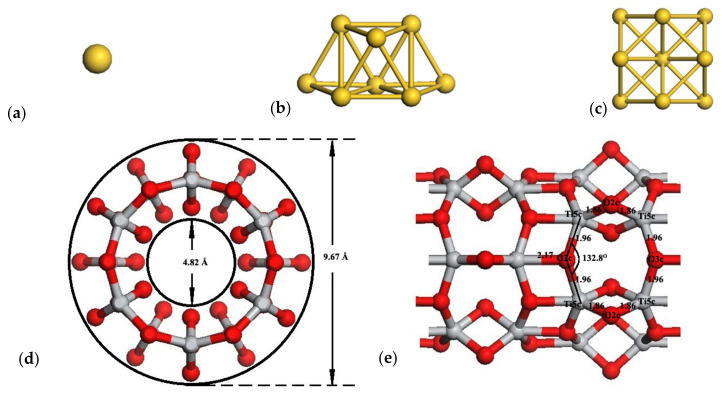
Optimized geometry of bare Au clusters and the bare TiO_2_NT (6,0). The first three configurations are: (**a**) Au_1_; (**b**) Au_8_biplanar; (**c**) Au_13_ cuboctahedral. (**d**,**e**) are the cross-sectional and the side view of the TiO_2_NT (6,0), respectively. The gold atoms are shown in gold, the oxygen atoms are displayed in red, while titanium atoms are presented in light gray.

**Figure 2 molecules-27-02756-f002:**
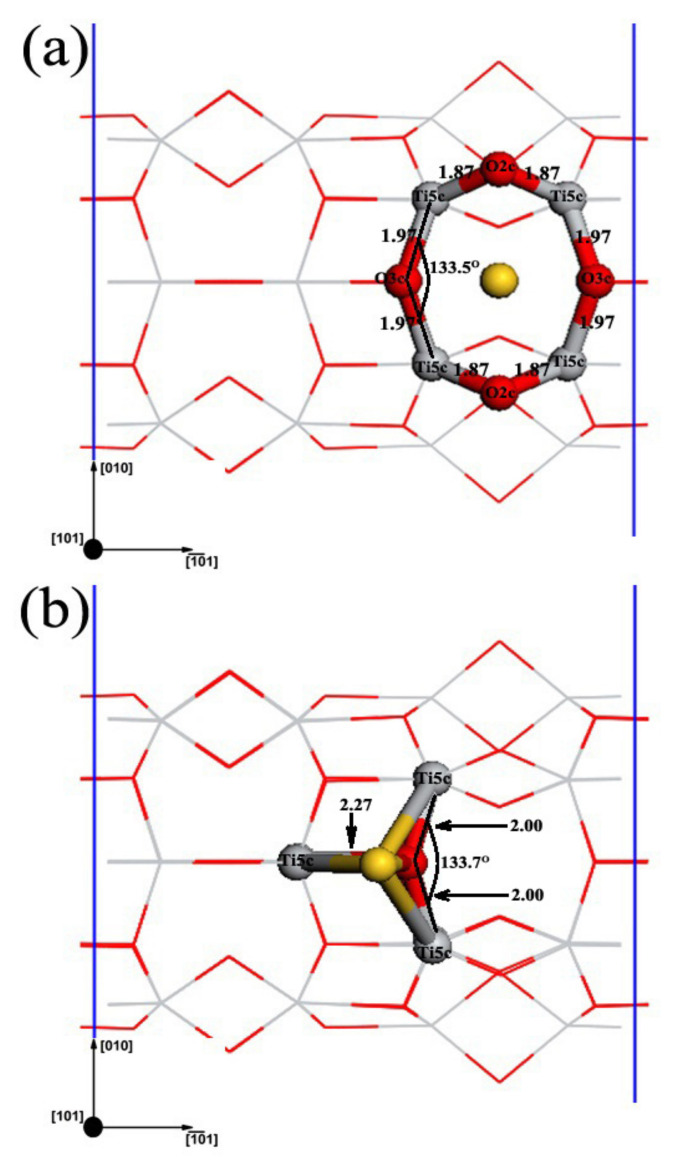
Top views of detailed adsorption structures for a single Au adatom on the TiO_2_NT (6,0) surface: (**a**) Au_1(O,O)_; (**b**) Au_1(O,Ti)_.

**Figure 3 molecules-27-02756-f003:**
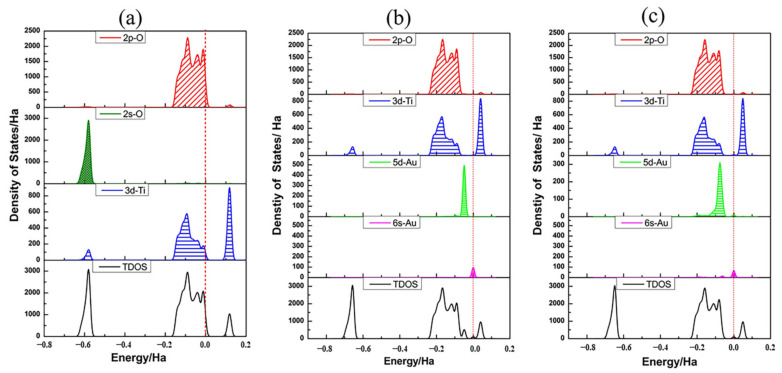
Density of state profiles and the partial density of states of TiO_2_NT (6,0) and Au and relevant nanotube surface atoms for a single Au adatom in two different adsorption states: (**a**) bare TiO_2_NT (6,0), (**b**) Au_1(O,O)_, (**c**) Au_1(O,Ti)_. The black curve is the sum of PDOS of O atoms, Ti atoms, and Au atoms from the optimized bare TiO_2_NT (6,0)/Au systems, while the red curve indicates the 2*p* orbital of oxygen atoms, the dark green curve indicates the 2*s* orbital of oxygen atoms, the blue curve represents the 3*d* orbital of titanium atoms, the light green curve denotes the 5*d* orbital of gold atoms, and the magenta curve represents the 5*s* orbital of gold atoms, respectively. The Fermi level of bare TiO_2_NTs is set at 0 eV, as denoted in the red dashed line.

**Figure 4 molecules-27-02756-f004:**
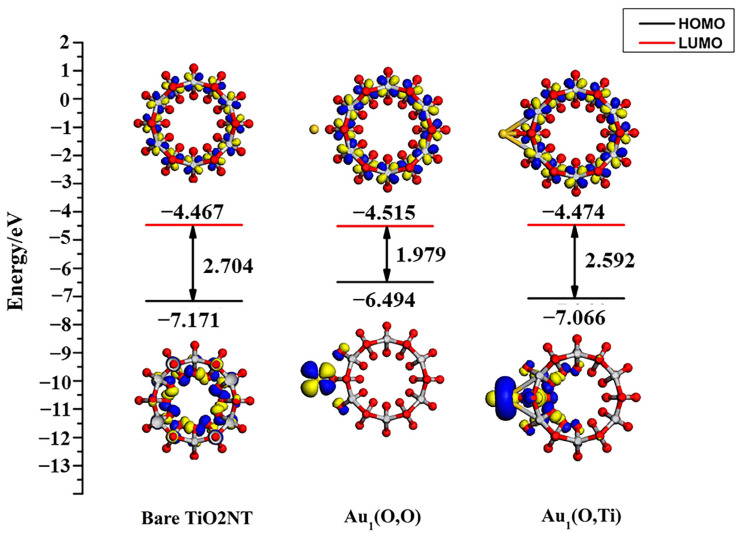
HOMO and LUMO along with the energy gap of bare TiO_2_NT and two Au_1_ adsorption systems calculated at the Γ-point. The isovalue set is 0.025 electrons/Å^3^. The gold atoms are shown in gold, the oxygen atoms are displayed in red, while titanium atoms are presented in light gray. Blue and yellow regions represent the positive and negative parts of HOMO and LUMO respectively.

**Figure 5 molecules-27-02756-f005:**
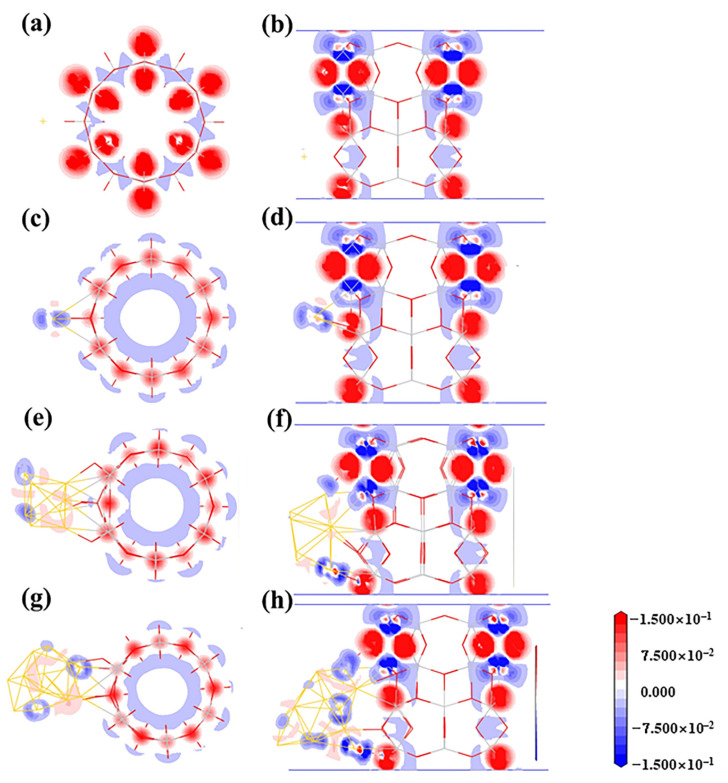
Deformation density contour maps for the adsorption of Au*_n_* clusters on the TiO_2_NT (6,0) surface. The configurations are as follows: (**a**,**c**,**e**,**g**) are the cross-sectional views of Au_1(O,O)_, Au_1(O,Ti)_, Au_8-C,_ and Au_13-B_, respectively; (**b**,**d**,**f**,**h**) are the side views of Au_1(O,O)_, Au_1(O,Ti)_, Au_8-C,_ and Au_13-B_, respectively. Contour level is −0.015~0.015 au. In this plot, a loss of electrons is indicated in blue, while electron enrichment is indicated in red.

**Figure 6 molecules-27-02756-f006:**
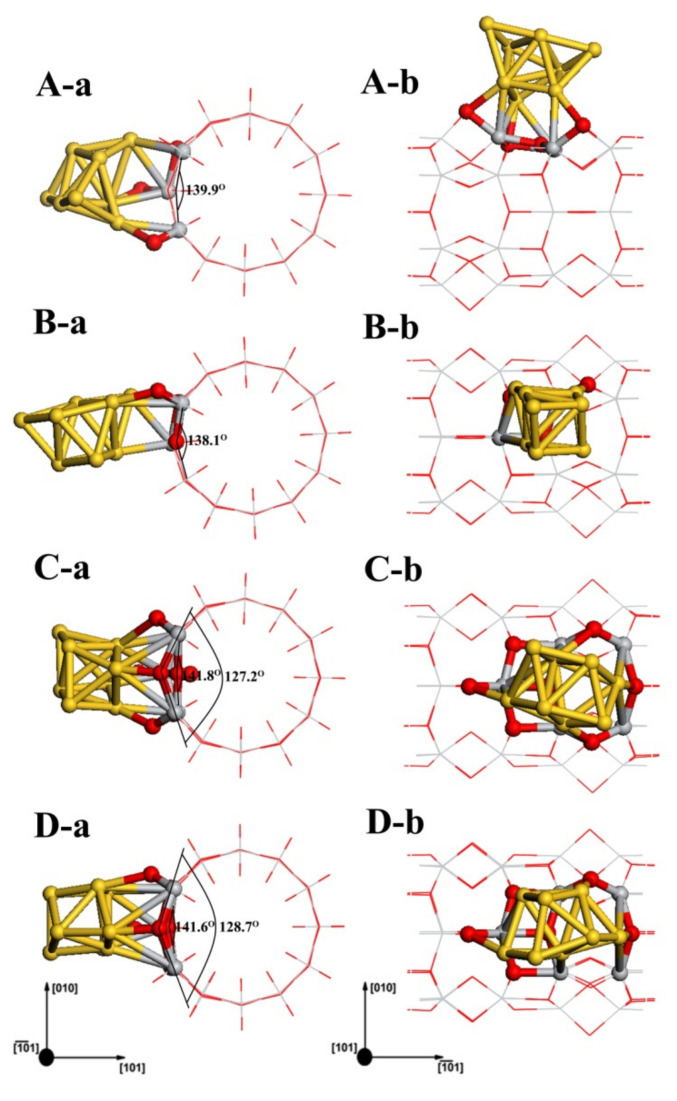
Four adsorption geometries for Au_8_ clusters on the TiO_2_NT (6,0) are displayed as (**A**–**D**). (**a**,**b**) depict the cross-sectional view and the side view of the Au_8_ cluster absorption systems, respectively. The cross-sectional views of four adsorption geometries are shown as (**A-a**,**B-a**,**C-a**,**D-a**), respectively. The side vies of four adsorption geometries are shown as (**A-b**,**B-b**,**C-b**,**D-b**), respectively.

**Figure 7 molecules-27-02756-f007:**
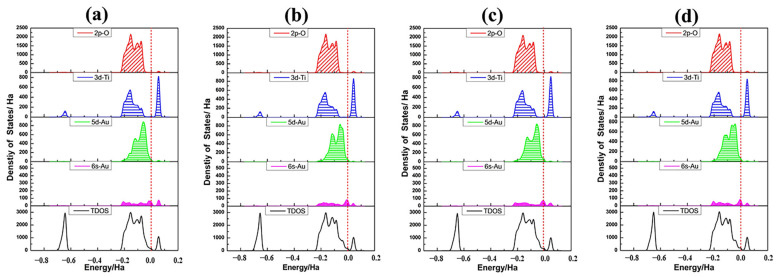
Density of state profiles and the partial density of states of Au_8_ and relevant nanotube surface atoms for Au_8_ clusters in four different adsorption states: (**a**) Au_8-A(2cO,3cO,2cO)_; (**b**) Au_8-B(2cO,5cTi)_; (**c**) Au_8-C(2cO,2cO,3cO)_; (**d**) Au_8-D(2cO,2cO,3cO)_. The detailed information is same as in Figure 3.

**Figure 8 molecules-27-02756-f008:**
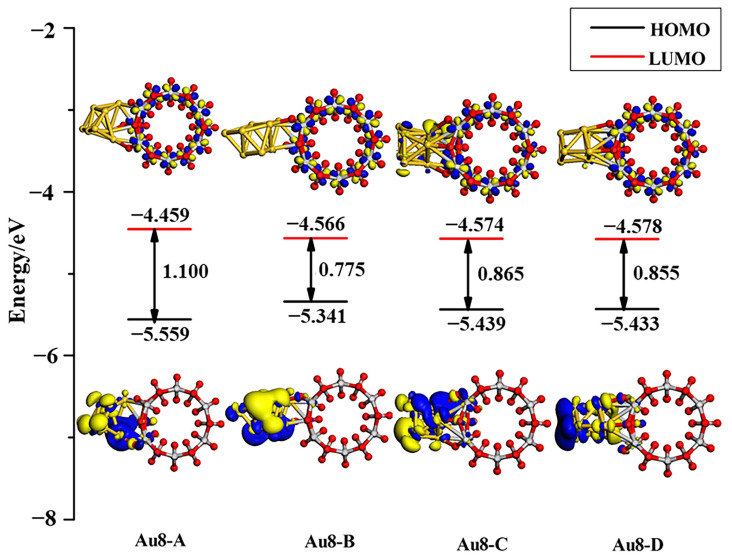
HOMO and LUMO along with the energy gap of four Au_8_ adsorption systems calculated at the Γ-point. The isovalue set is 0.025 electrons/Å^3^.

**Figure 9 molecules-27-02756-f009:**
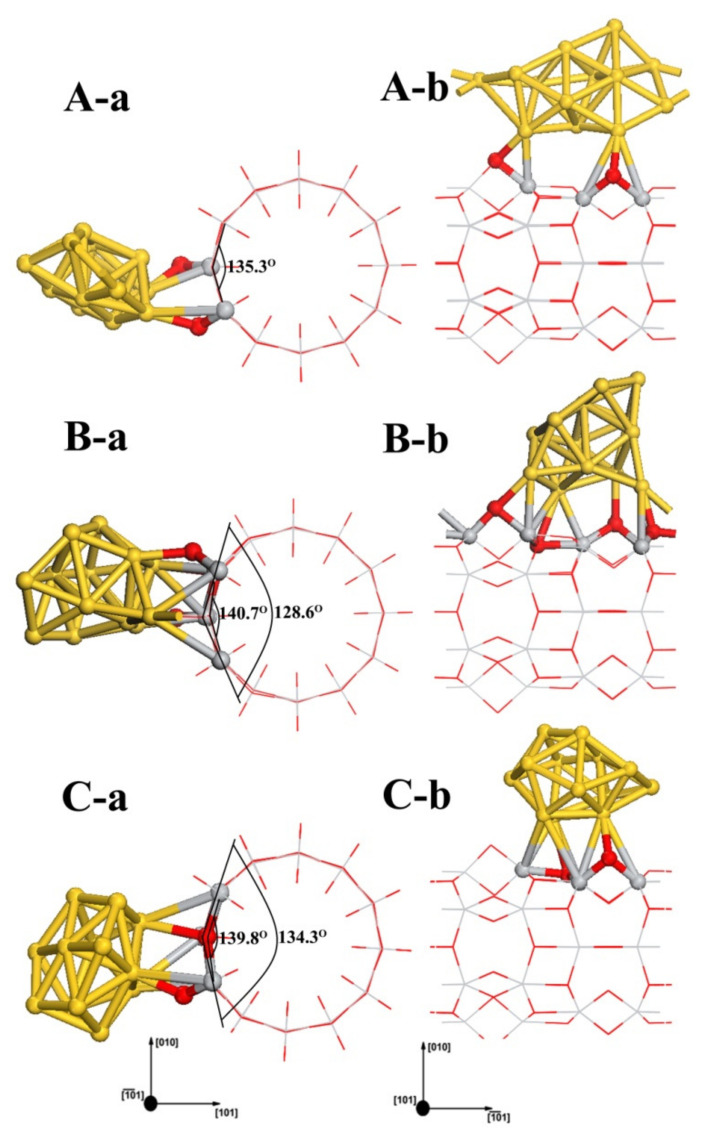
Four adsorption geometries for Au_13_ clusters on the TiO_2_NT (6,0) are displayed as (**A**–**C**). (**a**,**b**) depict the cross-sectional view and the side view of the Au_13_ cluster absorption systems, respectively. The cross-sectional views of four adsorption geometries are shown as (**A-a**,**B-a**,**C-a**), respectively. The side vies of four adsorption geometries are shown as (**A-b**,**B-b**,**C-b**), respectively.

**Figure 10 molecules-27-02756-f010:**
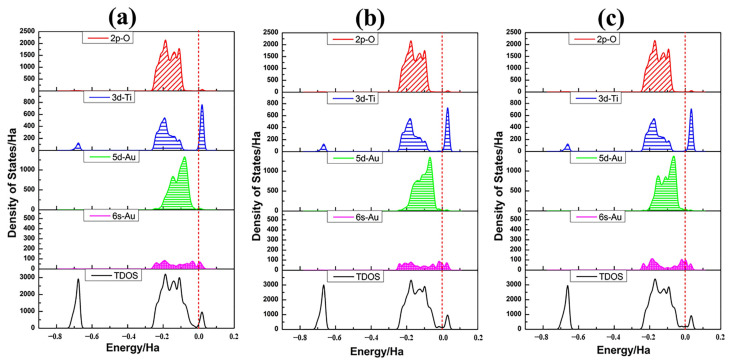
Density of state profiles and the partial density of states of Au_13_ and relevant nanotube surface atoms for Au_13_ clusters in four different adsorption states: (**a**) Au_13-A(2cO,2cO)_, (**b**) Au_13-B(2cO,3cO,2cO,3cO)_, (**c**) Au_13-C(3cO,2cO)_. The detailed information is the same as in Figure 3.

**Figure 11 molecules-27-02756-f011:**
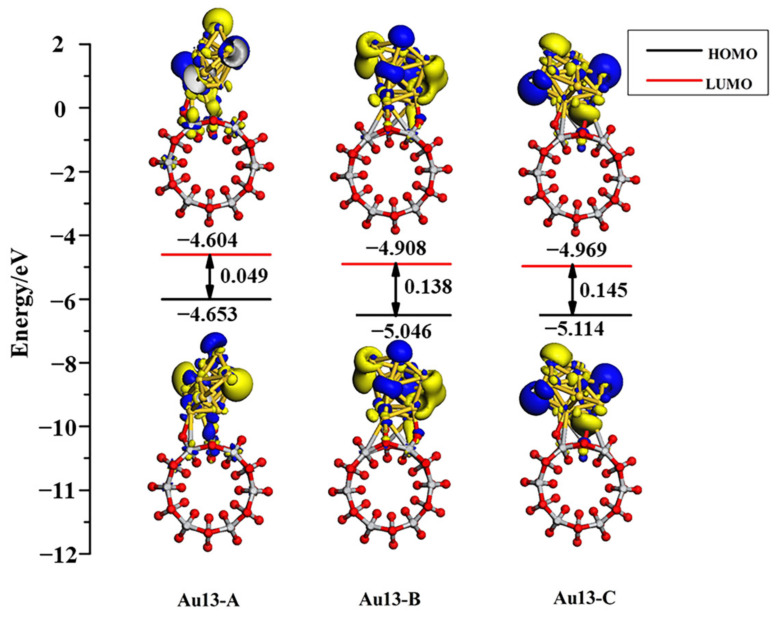
HOMO and LUMO along with the energy gap of four Au_13_ adsorption systems calculated at the Γ-point. The isovalue set is 0.025 electrons/Å^3^.

**Table 1 molecules-27-02756-t001:** Adsorption and clustering energies of Au_1_ and atomic Mulliken charges of Au and TiO_2_NT surface atoms for the Au_1_/TiO_2_NT system in different configurations ^a^.

Configuration	Energy	Mulliken Charge (a.u.)							
	$E_{Au_{1}}^{ads}(\mathrm{eV})$	Au	3cO ^1^	2cO ^1^	3cO ^2^	2cO ^2^	5cTi ^1^	5cTi ^2^	5cTi ^3^
Bare TiO_2_NT			−0.966	−0.779	−0.965	−0.779	1.714	1.714	1.713
Au_1(O,O)_	0.20	−0.060	−0.958	−0.770	−0.958	−0.770			
Au_1(O,Ti)_	0.49	−0.075			−0.940		1.732	1.731	1.676

^a^ The superscripts on O and Ti indicate to which Au atom they are bounded.

**Table 2 molecules-27-02756-t002:** Adsorption and clustering energies of Au_8_ and atomic Mulliken charges of Au and TiO_2_NT surface atoms for the Au_8_/TiO_2_NT system in different configurations ^a^.

Configuration	Energy					Mulliken Charge (a.u.)							
	$E_{Au_{8}}^{ads}(\mathrm{eV})$	$E_{Au_{8}}^{clu}(\mathrm{eV}/\mathrm{atom})$	Au_8_	3cO ^1^	2cO ^1^	2cO ^2^	3cO ^2^	2cO ^3^	5cTi ^1^	5cTi ^2^	5cTi ^3^	5cTi ^4^	5cTi ^5^
Bare TiO_2_NT				−0.966	−0.779	−0.779	−0.966	−0.779	1.717	1.714	1.718	1.714	1.713
Au_8-A(2cO, 3cO, 2cO)_	1.11	2.12	−0.321		−0.836		−0.965	−0.809	1.720	1.834		1.766	1.892
Au_8-B(2cO,5cTi)_	0.86	2.09	−0.293			−0.851	−0.960					1.815	1.803
Au_8-C(2cO,2cO,3cO)_	0.81	2.09	−0.424	−0.993	−0.826	−0.758		−0.843	1.926	1.874	1.716	1.783	1.905
Au_8-D(2cO,2cO,3cO)_	0.72	2.07	−0.343	−0.987		−0.818		−0.817	1.733	1.769	1.751	1.783	1.870

^a^ The superscripts on O and Ti indicate to which Au atom they are bound.

**Table 3 molecules-27-02756-t003:** Adsorption and clustering energies of Au_13_and atomic Mulliken charges of Au and TiO_2_NT surface atoms for the Au_13_/TiO_2_NT system in different configurations ^a^.

Configuration	Energy						Mulliken Charge (a.u.)								
	$E_{Au_{13}}^{ads}(\mathrm{eV})$	$E_{Au_{13}}^{clu}(\mathrm{eV}/\mathrm{atom})$	Au_13_	3cO ^1^	2cO ^1^	3cO ^2^	2cO ^2^	3cO ^3^	2cO ^3^	5cTi ^1^	5cTi ^2^	5cTi ^3^	5cTi ^4^	5cTi ^5^	5cTi ^6^
Bare TiO_2_NT				−0.966	−0.779	−0.965	−0.779	−0.966	−0.779	1.717	1.714	1.718	1.714	1.713	1.717
Au_13-A(2cO,2cO)_	3.12	2.28	−0.225		−0.870				−0.833	1.788	1.794			1.819	
Au_13-B(2cO,3cO,2cO,3cO)_	2.23	2.21	−0.322	−0.996			−0.809	−0.929	−0.830	1.736		1.761	1.805	1.848	1.727
Au_13-C(3cO,2cO)_	1.6	2.19	−0.171		−0.854	−0.934				1.762	1.791		1.737	1.778	

^a^ The superscripts on O and Ti indicate to which Au atom they are bound.

## Data Availability

Not applicable.

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
