# Peer review of "DFT Investigations of Aun Nano-Clusters Supported on TiO2 Nanotubes: Structures and Electronic Properties"

_molecules, 2022, doi:10.3390/molecules27092756_

Round 1
Reviewer 1 Report
In the present manuscript, Wang and Zhou perform a density-functional-theory-based study of Au nanoclusters adsorbed to a TiO2 nanotube substrate to investigate potential enhancement of photocatalytic and photoelectric effects. They examine different sizes of Au clusters (Au1, Au8 and Au13) and consider different adsorption sites in terms of their energetics, the electronic structure (total and partial DOS, as well as band gap sizes), atomic Mulliken charges and electron density deformation maps. The key findings that Au nanoclusters exhibit higher adsorption energies as a function of their size and that the most stable adsorption site configurations decrease the size of the band gap in comparison to the pure TiO2 nanotube (an effect that is traced back to the role of the 6s and 5d orbitals of Au) seem to suggest a potential application as an enhanced photoelectric material, which may certainly be of interest to the wider scientific community.
From a purely methodological point of view, the research is well-conceived, and the methodology is clear, albeit the description could be more detailed in places. The conclusions are backed up by the observed results and seem valid. However, there are a few points at this level that need to be addressed:
- In the methodological description, the authors describe their simulation cell as being “40 Å × 40 Å × 11 Å”, (line 121-122) with 11 Å being the value of the TiO2 nanotube length, yet simultaneously select a k-mesh of 4 × 4 × 2 (line 124). This seems like a weird choice given the dimensions of the supercell – I myself would have rather gone for something like 2×2×4 to ensure better convergence. Why has the mesh been chosen in this manner?
- The authors state: “Given the error bar imposed on our calculations of the nanotube-supported clusters by the finite size of the periodic unit of nanotube and by the neglect of spin polarization, we believe that the quality of our results for the studied system is quite acceptable” (lines 129-132). This seems like a rather circular argument: the methodology limits the accuracy due to several factors, and thus, the accuracy is good enough? How large is the introduced “error bar” exactly, i.e., what percentage of the total energy could we be talking about here? What would be the accuracy/computational cost-tradeoff for spin-polarized calculations?
- Line 133-134: The authors refer to dAP, which apparently is the Ti-O top bond length. That ought to be a known value – why not just provide it directly in the text?
- Figure 1: Please verify again that the caption and the actual subfigures match! I have very strong doubts that this is the case for c), d) and e), whereas f) and g) are not even mentioned in the caption.
- Line 190: When discussing Mulliken charges, the claim is that “5cTi” atoms became more positive” upon cluster adsorption. This is not the case for the 5cTi3 atom, though, as evident from Table 1.
- The discussion of band gaps via the HOMO/LUMO scheme is entirely valid; however, as the authors already go into detail while analyzing the DOS of the different clusters and their constituting species, the band gap can likewise be obtained from the band structure. This has the added advantage of making estimation whether the band gap is direct or indirect possible, which may be of relevance for photoelectric applications. Why was this not done in this manner?
- Conclusion, line 415: “The adsorption energy is increased as the number of Au atoms increased linearly” – it would be extremely helpful to plot this dependency in a dedicated figure, instead of giving the values in three different tables scattered through the whole manuscript (by the way, in Table 3, the subscript refers to “Au8”, not “Au13”.
These are the points that need to be addressed on the scientific side, which is, in general fairly sound, as said.
However, stylistically, unfortunately, the manuscript is an entirely different matter. While I realize that the authors are not native English speakers, I must, without any pleasure and regretfully, admit that this is the worst-written manuscript I have read in a long while. The amount of stylistic and grammatical mistakes is simply enormous and there are sentences which are entirely incomprehensible, such as:
“The research of nanostructure model of metal clusters supported on well-ordered metal oxide surfaces are signality“ (very first sentence, line 24-25 – what does “signality” even mean?)
or
“A 3D random network of nanoparticles with more particle boundaries than 1D-nanostructures when the transport of charge carriers.” (line 49-50), where the sentence seems to lack a meaningful verb
or
“For the projection of 2p-orbitals of oxygen, 3d-orbitals of titanium and 5d-orbitals of gold showed the major contribution to the PDOS in energy range of our interested” (line 202-204) – interested *what*?
These were just 3 examples of entirely incomprehensible sentences out of a much vaster reservoir of typos, mistakes and stylistically bad choices, which I cannot by any means list here in full, since there is something to list in approximately every third line. Also, the formatting in relation to subscripts/superscripts is shoddy. The entire manuscript needs to be rewritten to a better language standard and, for the sake of the authors’ themselves, checked by a native English speaker or a professional translator, if possible. In the present state, publication is out of the question.
With this being said and combining the points outlined above in the scientific part as well as the generally bad linguistic state, major revisions are needed before a reconsideration of publication can be made.
Reviewer 2 Report
1- Extensive editing of English language and style required.
2- "Introduction" section lacks the literature review over theoretical studies on AU clusters deposited on TiO2 (clusters/surfaces/NTs).
3- In "Methodology" section, you noted "we believe that the quality of our results for the studied system is quite acceptable [3]."!! for that claim you should present/explain your reasons.
4- What is the challenge for which you were looking? What are you scopes? They must clearly be mentioned in "Introduction" part.
5- To evaluate/verify your results, you should compare your results with the published/approved experimental data (e.g. your estimated band gap with the related experimental ones) or with previous theoretical results (e.g. geometries of AU nanoclusters on TiO2 surfaces).
Round 2
Reviewer 1 Report
In the second revision of the present manuscript, the authors addressed the remarks from the previous iteration satisfactorily. While the justification of choosing a k-mesh of a given size because a cited paper "did it that way" is not entirely pleasing, basing methodological choices on preceding work treating similar systems is valid and indicates that the methodology is suited. Likewise, I still would have liked to see a discussion of direct vs. indirect band gap character (not size) specifically by means of the band structure (not the DOS), which the authors, despite the statement in the cover letter, have not provided, but that is a rather minor point.
The other issues were addressed, Figure 1 was fixed and the writing, thanks to the editorial services of MDPI, was significantly improved. Thus, I have no objection to accepting and publishing the manuscript in its present, revised form.